# Impact of the First COVID-19 Wave on French Hospitalizations for Myocardial Infarction and Stroke: A Retrospective Cohort Study

**DOI:** 10.3390/biomedicines10102501

**Published:** 2022-10-07

**Authors:** Anne-Sophie Mariet, Gauthier Duloquin, Eric Benzenine, Adrien Roussot, Thibaut Pommier, Jean-Christophe Eicher, Laura Baptiste, Maurice Giroud, Yves Cottin, Yannick Béjot, Catherine Quantin

**Affiliations:** 1Biostatistics and Bioinformatics (DIM), University Hospital of Dijon, 21000 Dijon, France; 2Clinical Epidemiology/Clinical Trials Unit, Clinical Investigation Center, University Hospital of Dijon, CIC1432, 21000 Dijon, France; 3Neurology Department, University Hospital of Dijon, 21000 Dijon, France; 4Dijon Stroke Registry (Santé Publique France-Inserm), University of Burgundy, UFBC, 21000 Dijon, France; 5EA 7460 (Pathophysiology and Epidemiology of Cerebro-CardioVascular Diseases), University of Burgundy, UFBC, 21000 Dijon, France; 6Cardiology Department, University Hospital of Dijon, 21000 Dijon, France; 7Pathophysiology and Epidemiology of Cerebro-CardioVascular Diseases, University of Burgundy, 21000 Dijon, France; 8Registre des Infarctus du Myocarde de Côte d’Or, University Hospital of Dijon, 21000 Dijon, France; 9Université Paris-Saclay, UVSQ, University of Paris-Sud, Inserm, High-Dimensional Biostatistics for Drug Safety and Genomics, CESP, 94800 Villejuif, France

**Keywords:** stroke, transient ischemic attack, myocardial infarction, Coronavirus Disease 2019, lockdown, spatial autocorrelation

## Abstract

The COVID-19 pandemic modified the management of myocardial infarction (MI) and stroke. We aimed to evaluate the effect of the COVID-19 pandemic on the volume and spatial distribution of hospitalizations for MI and stroke, before, during and after the first nationwide lockdown in France in 2020, compared with 2019. Hospitalization data were extracted from the French National Discharge database. Patient’s characteristics were compared according to COVID-19 status. Changes in hospitalization rates over time were measured using interrupted time series analysis. Possible spatial patterns of over or under-hospitalization rates were investigated using Moran’s indices. We observed a rapid and significant drop in hospitalizations just before the beginning of the lockdown with a nadir at 36.5% for MI and 31.2% for stroke. Hospitalization volumes returned to those seen in 2019 four weeks after the end of the lockdown, except for MI, which rebounded excessively. Older age, male sex, elevated rate of hypertension, diabetes, obesity and mortality characterized COVID-19 patients. There was no evidence of a change in the spatial pattern of over- or under-hospitalization clusters over the three periods. After a steep drop, only MI showed a significant rebound after the first lockdown with no change in the spatial distribution of hospitalizations.

## 1. Introduction

The COVID-19 pandemic, which resulted in a long-term lockdown in most countries during the spring of 2020, modified the management of non-COVID-19 emergencies as myocardial infarction (MI), stroke and Transient Ischemic Attack (TIA) that are severe conditions needing emergency care, leading to a risk of death or motor and cognitive disturbances. Any delay in management of MI and stroke/TIA can have devastating adverse consequences on survival or functional outcome. 

Although an association was found between SARS-CoV-2 infection and the occurrence of MI or stroke [1,2], several studies have observed a steep decline in their hospital admissions for MI and stroke [3,4,5,6,7,8,9,10,11,12]. This decrease of hospital admissions during the lockdown was associated with an increase in mortality caused by ischemic heart disease in USA [13]. French studies found an increase in in-hospital mortality after admission for stroke [6] and MI [10,11] during the lockdown. However, the analysis of data did not provide sufficient information on the recovery of hospital admissions especially over a long period of time and at a fine spatial scale. Indeed, a significant decrease in hospital admissions for MI and stroke was observed during the first wave of COVID-19 in nine hospitals in the United Kingdom [3], for MI in Northern California [4], for stroke in a study including at least two stroke centers per country in 17 countries [5]. In France, a decrease in emergency department admissions for MI and stroke was found during the lockdown with regional disparities and a return to usual levels after the lockdown with an end of study on 31 May 2020 [8]. Other French studies found a decrease in hospital admissions during the lockdown for MI or stroke, but they did not cover the whole country [7,10,12], or had a short post-lockdown period with an end of study in April [7,12] or May [8] or June 2020 [6,9], or a large regional [7,8,11,12] or departmental spatial scale [9].

The primary objective of this study was to compare the volume of hospitalizations in France, irrespective of the size or type of hospital, before and during the first lockdown in spring 2020 for MI, acute stroke and TIA. The secondary objectives evaluated these conditions recovery volumes during a long post-lockdown period and to compare the clinical characteristics of patients with and without COVID-19, as well as to search possible spatial patterns of over or under-hospitalization rates during these three periods.

We hypothesized that different profiles in the drop and in the recovery of hospitalization volumes of these diseases would be observed, despite similar vascular profiles, reflecting different healthcare networks.

## 2. Materials and Methods

The nationwide data were provided by the French National Hospital Discharge database (Programme de Médicalisation des Systèmes d’Information, PMSI), and it was approved by the National Committee for Data Protection and by the Agency for Information on Hospital Care (Agence Technique de l’Information sur l’Hospitalisation, ATIH).

### 2.1. Hospitalization Data

Hospitalization data from January to September 2019 and 2020 were extracted from the French National Discharge database. According to the International Classification of Diseases-Tenth Revision (ICD-10), the codes were I21, I22, I23; I214; and I21, I22, I23 but not I214 for MI, Non-ST segment elevation myocardial infarction (NSTEMI), and ST segment elevation myocardial elevation (STEMI), respectively. Cerebrovascular events included overall stroke, ischemic stroke (IS), hemorrhagic stroke (HS) and TIA. They were identified according to the International Classification of Diseases-Tenth Revision (ICD-10) codes: the codes were I60, I61, I629; I63 and I64; and G45 for HS, IS and TIA, respectively. The codes were considered as a primary diagnosis but also as associated and secondary diagnoses to ensure that these diseases were identified even if another severe disease was the primary diagnosis. COVID-19 was identified using specific codes created by ATIH. Other variables were extracted: age, sex and available vascular risk factors (hypertension, diabetes, obesity, atrial fibrillation (AF)).

### 2.2. Study Design

We retrospectively analyzed all patients admitted to every public and private French hospital for MI, acute stroke and TIA, between January and September 2020. This period included the first peak of the COVID-19 pandemic and the first national lockdown from 17 March to 10 May 2020. Hospitalization and in-hospital death rates were compared with the rates from the same period during 2019 week per week. All data were anonymous. This study was authorized by the French Data Protection Authority on 3 July 2020 (Registration number: DR-2020-250 on 7 March 2020.

### 2.3. Statistical Analysis

Qualitative variables are presented as frequencies (percentage). Quantitative variables are presented as means ± standard deviation. The different variables analyzed in the cohort of patients with stroke and MI were compared using the Chi-2 test or the Fisher’s exact test (for qualitative variables) and Student’s *t*-test or Mann–Whitney test (for quantitative variables) according to COVID-19 status.

Standardized rates of hospitalizations for each condition were calculated on European and world population per 100,000 person-months for the three periods: before, during and after the lockdown.

Interrupted time series analysis was performed to measure changes in hospitalizations rates over time for each condition in January to September 2020 with three periods before (1 January to 16 March 2020), during (17 March to 10 May 2020) and after the lockdown (11 May to 30 September 2020). This model used weekly hospitalizations rates over the study period and included a linear time trend. We thus quantified the impact of the lockdown as changes in the level and slope compared with the preceding period. The change in the number of stays for each condition in 2020 compared with 2019 by week was plotted as smoothed curves using degree 2 spline functions.

Global standardized hospitalization rates for stroke and MI were calculated according to the French population structure and were mapped at the scale of zip-codes. The spatial distribution of rates for the three studied periods was represented on maps.

We assessed the overall autocorrelation of the spatial distribution of stroke and MI hospitalization rates using a univariate Moran’s I test. Possible local spatial autocorrelations were evaluated using local spatial measures of association (LISA: Local Indicators of Spatial Association), which allow us to verify the existence of spatial clusters of over- or under-hospitalization and to visualize their location.

The statistical significance threshold was set at <0.05. Spatial contiguity for spatial autocorrelation tests was assessed as Queen contiguity, their significance was assessed by comparison to a reference distribution using 999 random permutations.

All statistical analyses were performed using SAS (SAS Institute Inc., Version 9.4, Cary, NC, USA). Spatial autocorrelation analyses were conducted using Geoda software (Geoda 1.20., Center for Spatial Data Science – University of Chicago (IL, USA) [1]).

## 3. Results

For MI, from 1 January 2019 to 30 September 2019 and 1 January 2020 to 30 September 2020, respectively, 88,714 and 84,318 cases were reported, which was a 5.0% decrease. There were 46,764 NSTEMI in 2019 versus 44,719 in 2020 (decrease of 4.4%), while there were 47,193 STEMI in 2019 versus 44,424 in 2020 (decrease of 5.9%). In 2020, the number of hospital admissions for MI decreased between week 10 and 20, with a nadir in week 13 and a 36.5% drop compared to 2019 (Figure 1).

There was a major drop in the NSTEMI group (41.3%), compared with the STEMI group who saw a drop of 32.9% in week 13. After the end of the lockdown, the number of MI, NSTEMI and STEMI reached prior levels on week 22. Subsequently, there was a significant increase of the volume of cases from week 23 to week 33 for STEMI, and 5.40% more cases from week 24 to week 33 for NSTEMI (95% CI: 4.97–5.82%) compared with 2019. During the lockdown, COVID-19 infection was identified in 4.7% of overall MI, 5.1% of STEMI, and 4.5% of NSTEMI (Appendix A).

For cerebrovascular events, during the first period (1 January 2019, to 30 September 2019) and the second period (1 January 2020 to 30 September 2020), 146,511 and 138,918 events were reported, respectively, showing a decrease of 5.2% in admissions. The number of IS was 91,037 in 2019 versus 86,569 in 2020, while HS were 27,729 in 2019 versus 26,516 in 2020. The proportion of the decrease in admissions was similar for IS (4.9%) and HS (4.4%). In addition, 36,882 TIA patients were admitted to hospital in 2019 and 34,219 in 2020, equaling a drop in admissions of 7.2% in 2020. During the first lockdown in 2020, the number of overall stroke admissions decreased suddenly just before the beginning of the lockdown (Figure 2), reaching a nadir in week 12, with a 31.2% drop in overall stroke admission in parallel with the peak of COVID-19 pandemic compared to 2019. A similar drop was observed for IS, with 26.1% fewer admissions in week 12, and HS with 26.1% fewer admissions in week 12. From the end of the lockdown until weeks 23–24, the number of stroke admissions gradually returned the levels seen in 2019. The number of TIA patients also decreased, with a nadir at week 12 and a recovery towards the levels of 2019 by week 23 (Figure 2).

At 43.9%, the drop was particularly steep. All of the figures display the usual decline in admissions related to the holiday effect in August (weeks 31 to 33). During the lockdown, COVID-19 was identified in 5.6% of overall stroke cases, 5.9% of IS, 6.8% of HS and 3.9% of TIA (Appendix A).

European and world-standardized rates of hospitalizations before, during and after the lockdown for each condition are available in the Appendix A allowing for international comparisons.

Using interrupted time series for 2020, we observe for all conditions, a significant decrease in level during compared with before lockdown, a significant increasing slope during lockdown, and a significant decreasing slope after lockdown (Appendix A). We also observe a significant decreasing slope before lockdown for ischemic stroke and NSTEMI, a significant increase in level after compared with during lockdown for STEMI (*p* = 0.037).

Patients with COVID-19 were older for MI, NSTEMI and STEMI (Table 1). Mean age did not differ between patients with and without COVID-19 for all strokes and for IS, while patients with COVID-19 were significantly older for TIA and younger for HS (Table 2).

Patients with COVID-19 were more frequently male when compared with non-COVID-19 for all strokes, IS and HS. Hypertension was more frequent in COVID-19 patients for MI, NSTEMI, STEMI and TIA while obesity was more frequent in COVID-19 patients for STEMI, all strokes and HS. AF was more predominant for MI, NSTEMI, STEMI and TIA. In-hospital mortality rates were significantly higher in COVID-19 patients for both MI (Table 1) and cerebrovascular events (Table 2).

The mapping of hospitalization rates did not show any spatial structuring of clusters of over- or under-hospitalization for stroke rates in France during the three periods, except a slightly cluster of over-hospitalization rate for MI in the northeast of the country, which persisted over the three periods studied (Figure 3 and Figure 4).

This result was confirmed by the Moran’s I statistics (Table Moran’s I by period and condition), which ranged from 0.0189 to 0.0278 for hospitalization rate for stroke during the studied period and from 0.0253 to 0.0743 for hospitalization rate for MI. These results suggest a poor spatial autocorrelation, whatever the condition or the period considered.

## 4. Discussion

To the best of our knowledge, this is the first nationwide population-based retrospective cohort study to compare hospital admissions in all public and private hospitals for STEMI, NSTEMI, IS, HS and TIA, before, during and after the end of the first lockdown implemented in France in 2020 with a post-lockdown period as long as 20 weeks. The magnitude was specific to each disease with a deep drop for NSTEMI (41.3%) and TIA (43.9%). The original information provided by this work is the description of the speed of the recovery of cases that was progressive for stroke before the end of the lockdown and which returned to admission rates similar to 2019, 4 weeks after the end of the lockdown. A major difference was observed for MI, STEMI and NSTEMI, which experienced a significant increase of 5.40% for NSTEMI over the levels observed in 2019.

For MI [2,3,4,5,6,7] and overall stroke [8,9,10,11,12,13], a drop in admissions by 30% was recorded in developed countries during the lockdown, especially in elderly patients [4,11]. Several studies reported also that the deepest drop was observed for NSTEMI [4,5,6,7] and TIA [10,11]. The reduction in hospital admissions for MI, stroke and TIA likely has common causes. The health authorities focused heavily on the message that individuals should stay home if they were experiencing mild symptoms to avoid the risk of COVID-19 contamination in hospital [13,14]. Priority was given to patients with COVID-19, which may have penalized many other emergencies such as MI and stroke [5,6,7,9,10,13]. Patients may have been afraid of exposure to COVID-19 in the hospital, especially the elderly, who may have stayed at home with their symptoms without seeking medical attention, resulting in their admission to the emergency department in a more severe condition than the younger population [15]. A rise in out-of-hospital cardiac arrests leading to death may explain the deep drop in MI [6]. Social isolation, misdiagnosis in emergency rooms are other possible explanations [6]. In addition, air pollution, which is a powerful risk factor for MI and stroke, dropped dramatically during the lockdown. The smaller drop observed for STEMI admissions compared with NSTEMI admissions could be the consequence of misdiagnosis (confusion with myocarditis), or due to the fact that pain is usually more intense in STEMI [6]. The main consequences for MI and stroke were the loss of opportunities to receive curative treatments and appropriate secondary prevention and to reduce preventable complications and excess deaths [16]. In fact, for stroke in France, a study found increases of both in-hospital and out-of-hospital 30-day case fatality rates for stroke during the first weeks of the lockdown [8]. However, a multicenter cohort study found that the quality of acute stroke care service did not change relevantly between the 2020 and 2019 period in Europe [17]. For MI, an increase in the symptom-onset-to-first-medical-contact time was found without increase of in-hospital mortality between 2020 and 2019 [4,18].

The long recovery period after the end of the lockdown comparing nationwide data on MI and stroke, which has not yet been described in the literature except for ischemic stroke only in Germany [19] may reflect several mechanisms: a decrease in the number of hospitalized COVID-19 patients, a change in patient behavior when experiencing the first symptoms of stroke and MI, an appropriate reactivation of emergency networks, and more available hospital facilities for non-COVID-19 emergencies. The lack of over-recovery for stroke sub-types observed also in the literature [19], contrasts with the higher significant recovery volume of MI, STEMI and NSTEMI volume could be explained by the fact that patients who did not access hospital care during the acute stage were admitted to the hospital later on for coronary angiography or mechanical treatments. To the contrary, the absence of over-recovery for stroke suggests that these patients were either lost to follow-up, were not called for another check-up, died at home, were unable to obtain medication during and after the lockdown, or were victim by psychological distress. These differences emphasize the weight of chest pain in emergency admissions. However, as COVID-19 resurged in 2020–2021, many countries have reimplemented lockdowns leading to a second decline of MI and stroke hospitalizations [20]. These trends may reflect a nationwide “learning effect” over time as described by Dengler et al. [19].

While our results suggest significant variation in hospitalization rates during the different periods following the onset of COVID-19 pandemic compared to the previous year, these differences do not appear spatially. Indeed, the spatial analysis did not show the emergence of significant clusters of over- or under-hospitalization during the period. The mapping of hospitalization rates did not show a clear spatial pattern, particularly for stroke. According to Moran’s indices, there was no evidence of spatial autocorrelation during or between the three studied periods, although there were a number of zip codes with high hospitalization rates along a northeast/southwest diagonal. This finding is consistent with previous work that showed that this area had higher hospital prevalence and mortality rates than the rest of the country [21,22].

On the one hand, it appears that the spatial distribution of hospitalization rates follows profound logics, which the pandemic did not modify. This spatial structuring seems to follow more individual behavioral (dietary habits, for example) or socio-geographic logics. Thus, numerous studies have shown the impact of the socio-residential environment on the spatial distribution of stroke or MI [23]. Moreover, mapping different pathologies that share common risk factors such as MI and stroke thus provides a better understanding of their spatial distribution and the at-risk areas in France [24].

However, on the other hand, this study also showed that there was a temporal variation as shown by the changes in hospital care supply brought about by the first months of the pandemic, which we found in the strong variation of hospitalization rates over time.

The effect of COVID-19 on the clinical characteristics of cardiac events and cerebral events are better known. The prevalence of diagnosed COVID-19 during the lockdown in all MI (4.7%) and in all stroke (5.6%) was similar to previous reports (1% to 6%) [25,26]. In addition, only in stroke, patients with COVID-19 were significantly older than non-COVID patients [27], except in HS, which is classical. In overall stroke, IS, HS and TIA, the proportion of men was greater in patients with COVID-19 than without COVID-19, as observed in previous work [14], in accordance with a greater risk of vascular complications in men than in women with COVID-19 [27]. In overall MI, NSTEMI and STEMI, the proportion of prior hypertension was significantly greater in patients with COVID-19 which is consistent with the fact that hypertension is involved in COVID-19 [28,29]. The proportion of diabetes was significantly greater in all of the conditions studied here in patients with COVID-19 as already reported [25,27]. By contrast, AF was predominant in COVID-19 patients associated with TIA [27], and in overall MI, NSTEMI, STEMI as a consequence [30] that could explain the excess of cardio-embolic ischemic stroke [27,30,31].

In line with the reported literature, in-hospital mortality in COVID-19 patients was significantly greater in MI [6,32], in overall stroke and TIA [10,25,26,33,34], consistent with the finding that COVID-19 and the presence of co-morbidities are a factor of excess mortality for stroke and MI [35]. The more frequent COVID-related death among the patients with MI or stroke could also explain the decrease in hospital admissions for MI and stroke if the death occurred at home.

The mechanisms of MI, stroke and TIA could be related to the pre-existing cardiovascular risk profile mainly in the elderly [31,35,36,37]. COVID-19 is known to induce a hypercoagulable state [27] and arrhythmia triggered by the infection-related inflammatory storm [28,30,32]. The higher prevalence of COVID-19 in HS patients raises questions about the impact of inflammation on the cerebral vessels [27], and the possible role of anticoagulants often used in COVID-19 [27,29,36]. However, MI and stroke can also be the result of COVID-19 itself [28,30,38,39,40,41], mainly affecting younger patients [39].

Limitations: This nationwide study is a descriptive, retrospective, observational study. Access times to hospital and revascularization procedures were not ascertained. Possible misclassification may have occurred. Under-detection is possible but likely minimal thanks to the long post-lockdown study period, resulting in complete data capture.

Strengths: We obtained all French hospital data from all public and private hospitals, and we were thus able to be nationally representative. The post-lockdown period was evaluated, and the possible risk of a seasonal effect was limited by using data from 2019.

## 5. Conclusions

We provide the first description of the fluctuations in French hospitalization volumes observed before, during and long after the first lockdown for COVID-19, including more than 84,000 MI and more than 138,000 stroke patient admissions. We observed a severe drop in hospital admissions during the lockdown followed by a rapid recovery to prior levels, except for MI, which exceeded prior levels, reflecting different management that is of great concern and suggesting a nationwide “learning effect” over time. The steep drop in admissions also suggests that many patients accessed neither acute care nor secondary prevention with severe consequences. The absence of spatial changes for the two conditions during the three periods studied seems to show that the pandemic had no effect on the geographical logics of their spatial distribution.

Therefore, it will be necessary to follow the trends of this dynamic to identify its causes between a simple effect of a brief recovery of the hospitalizations after the end of the lockdown, and a deep and long-lasting mutation of the non-COVID-19 emergencies practices boosted by the change in the behavior of the public and the health professionals.

## Figures and Tables

**Figure 1 biomedicines-10-02501-f001:**
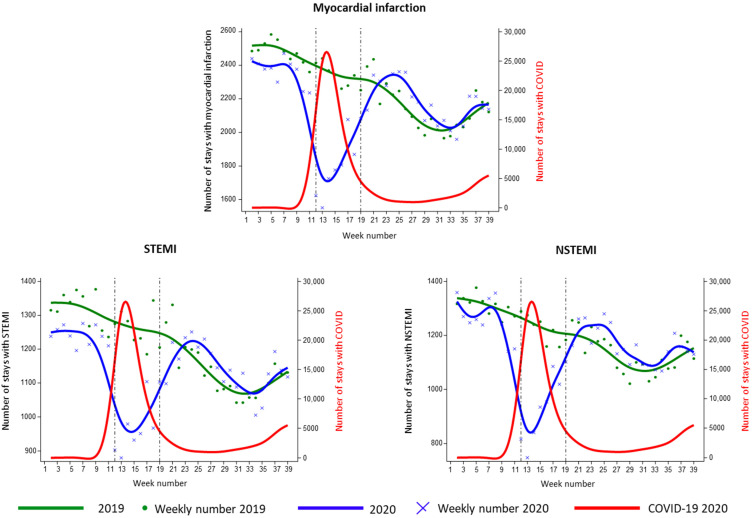
Hospital admissions for myocardial infarction and its subtypes in January to September 2020 versus 2019.

**Figure 2 biomedicines-10-02501-f002:**
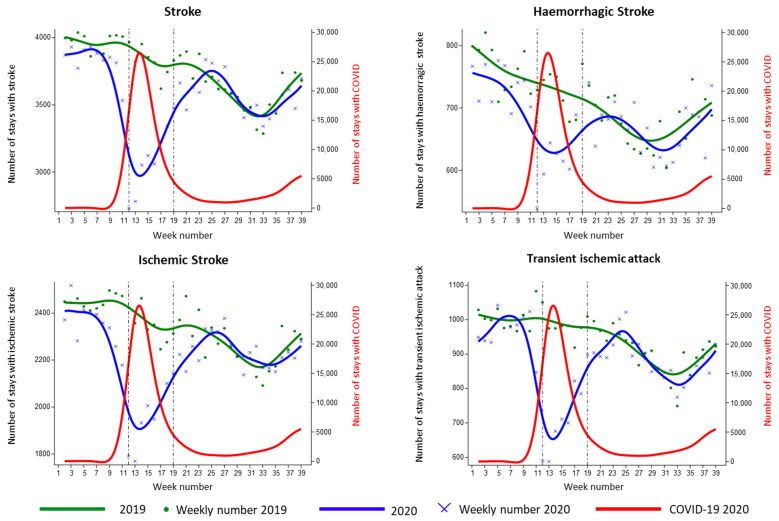
Hospital admissions for stroke and transient ischemic attacks in January to September 2020 versus 2019.

**Figure 3 biomedicines-10-02501-f003:**
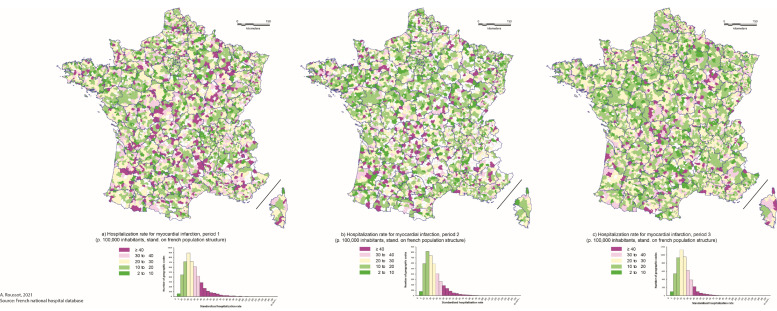
Hospitalization rate for myocardial infarction by periods and postal zip-codes in 2020. Legend: Hospitalization rates are standardized on French population structure and expressed per 100,000 inhabitants. Periods are defined as: (**a**) period 1 = before the lockdown (1 January to 16 March 2020), (**b**) period 2 = during the lockdown (17 March to 10 May 2020) and (**c**) period 3 = after the lockdown (11 May to 30 September 2020).

**Figure 4 biomedicines-10-02501-f004:**
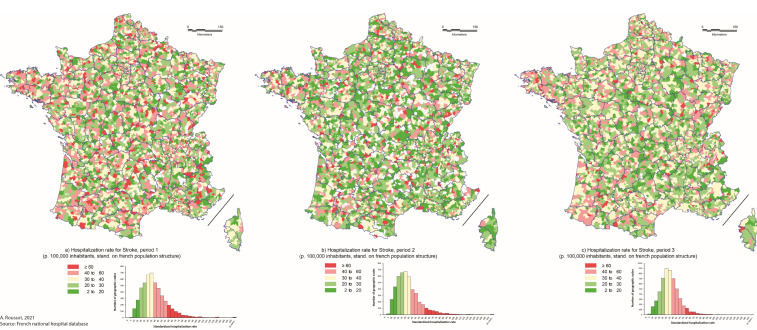
Hospitalization rate for stroke by periods and postal zip-codes in 2020. Legend: Hospitalization rates are standardized on French population structure and expressed per 100,000 inhabitants. Periods are defined as: (**a**) period 1 = before the lockdown (1 January to 16 March 2020), (**b**) period 2 = during the lockdown (17 March to 10 May 2020) and (**c**) period 3 = after the lockdown (11 May to 30 September 2020).

**Table 1 biomedicines-10-02501-t001:** Clinical characteristics of myocardial infarctions among COVID-19 vs. non-COVID-19 patients during the lockdown of 2020.

	COVID-19 + N (%) (n = 666)	COVID-19—N (%) (n = 13,536)	*p*-Value *
All MI (n = 14,202)			
Age, years			
Mean (SD)	73.0 (13.8)	68.8 (14.1)	<0.0001
Sex			0.10
Women	235 (35.3)	4367 (32.3)	
Men	431 (64.7)	9169 (67.7)	
Hypertension	356 (53.5)	5761 (42.6)	<0.0001
Diabetes	212 (31.8)	2993 (22.1)	<0.0001
Obesity	94 (14.1)	1582 (11.7)	0.058
Atrial fibrillation	169 (25.4)	2014 (14.9)	<0.0001
In-hospital mortality	183 (27.5)	979 (7.2)	<0.0001
NSTEMI (n = 7323)	327	6996	
Age, years			
Mean (SD)	73.7 (13.6)	70.3 (13.5)	<0.0001
Sex			0.93
Women	110 (33.6)	2369 (33.9)	
Men	217 (66.4)	4627 (66.1)	
Hypertension	201 (61.5)	3387 (48.4)	<0.0001
Diabetes	114 (34.9)	1758 (25.1)	<0.0001
Obesity	38 (11.6)	834 (11.9)	0.87
Atrial fibrillation	88 (26.9)	1091 (15.6)	<0.0001
In-hospital mortality	72 (22.0)	268 (3.8)	<0.0001
STEMI (n = 7663)	387	7276	
Age, years			
Mean (SD)	72.3 (13.7)	67.4 (14.5)	<0.0001
Sex			0.077
Women	135 (34.9)	2228 (30.6)	
Men	252 (65.1)	5048 (69.4)	
Hypertension	183 (47.3)	2762 (38.0)	0.0002
Diabetes	112 (28.9)	1419 (19.5)	<0.0001
Obesity	66 (17.1)	854 (11.7)	0.0017
Atrial fibrillation	96 (24.8)	1032 (14.2)	<0.0001
In-hospital mortality	121 (31.3)	753 (10.4)	<0.0001

N: number; %: percentage; SD: standard deviation; MI: myocardial infarction; NSTEMI: Non-ST segment elevation myocardial infarction; STEMI: ST segment elevation myocardial elevation. * *p*-value of the Chi-2 test or the Fisher’s exact test (for qualitative variables) and Student’s *t*-test or Mann–Whitney test (for quantitative variables).

**Table 2 biomedicines-10-02501-t002:** Clinical characteristics of cerebrovascular events among COVID-19 vs. non-COVID-19 patients during the lockdown of 2020.

	COVID-19 + N (%) (n = 1361)	COVID-19—N (%) (n = 22,995)	*p*-Value *
All stroke (n = 24,356)			
Age, years			
Mean (SD)	73.9 (14.5)	73.2 (14.8)	0.090
Sex			<0.0001
Women	566 (41.6)	11,178 (48.6)	
Men	795 (58.4)	11,817 (51.4)	
Hypertension	714 (52.5)	11,552 (50.2)	0.11
Diabetes	351 (25.8)	4274 (18.6)	<0.0001
Obesity	139 (10.2)	1734 (7.5)	0.0003
Atrial fibrillation	333 (24.5)	5110 (22.2)	0.053
In-hospital mortality	424 (31.2)	2811 (12.2)	<0.0001
Ischemic stroke (n = 15,411)	914	14,497	
Age, years			
Mean (SD)	74.4 (14.0)	74.2 (14.3)	0.76
Sex			<0.0001
Women	371 (40.6)	6869 (47.4)	
Men	543 (59.4)	7628 (52.6)	
Hypertension	489 (53.5)	7758 (53.5)	0.99
Diabetes	258 (28.2)	3092 (21.3)	<0.0001
Obesity	93 (10.2)	1248 (8.6)	0.10
Atrial fibrillation	252 (27.6)	3849 (26.6)	0.50
In-hospital mortality	274 (30.0)	1602 (11.1)	<0.0001
Hemorrhagic stroke (n = 4892)	335	4557	
Age, years			
Mean (SD)	68.6 (15.2)	70.7 (15.8)	0.023
Sex			0.0002
Women	125 (37.3)	2180 (47.8)	
Men	210 (62.7)	2377 (52.2)	
Hypertension	157 (46.9)	2191 (48.1)	0.67
Diabetes	74 (22.1)	638 (14.0)	<0.0001
Obesity	41 (12.2)	321 (7.0)	0·0005
Atrial fibrillation	63 (18.8)	949 (20.8)	0.38
In-hospital mortality	156 (46.6)	1373 (30.1)	<0.0001
TIA (n = 5673)	220	5453	
Age, years			
Mean (SD)	77.8 (14.9)	72.0 (14.7)	<0.0001
Sex			0.43
Women	108 (49.1)	2826 (51.8)	
Men	112 (50.9)	2627 (48.2)	
Hypertension	130 (59.1)	2444 (44.8)	<0.0001
Diabetes	47 (21.4)	841 (15.4)	0.017
Obesity	16 (7.3)	333 (6.1)	0.48
Atrial fibrillation	51 (23.2)	753 (13.8)	<0.0001
In-hospital mortality	31 (14.1)	86 (1.6)	<0.0001

N: number; %: percentage; SD: standard deviation; TIA: transient ischemic attack. * *p*-value of the Chi-2 test or the Fisher’s exact test (for qualitative variables) and Student’s *t*-test or Mann–Whitney test (for quantitative variables).

## Data Availability

The PMSI database was transmitted by the national agency for the management of hospitalization data. The use of these data by our department was approved by the National Committee for data protection. We are not allowed to transmit these data. PMSI data are available for researchers who meet the criteria for access to these French confidential data (this access is submitted to the approval of the National Committee for data protection) from the national agency for the management of hospitalization (ATIH—Agence technique de l’information sur l’hospitalisation). Address: Agence technique de l’information sur l’hospitalisation, 117 boulevard Marius Vivier Merle, 69329 Lyon Cedex 03.

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
