# Peer review of "Impact of the First COVID-19 Wave on French Hospitalizations for Myocardial Infarction and Stroke: A Retrospective Cohort Study"

_biomedicines, 2022, doi:10.3390/biomedicines10102501_

Round 1
Reviewer 1 Report
Please answer my question (see the text) and comment on the two mentioned issues.

Author Response
Please answer my question (see the text) and comment on the two mentioned issues.
- We thank the Reviewer for giving us an opportunity to substantially improve the content of our manuscript. We have modified the article according to your requests. You will find the proposed modifications in the text using track changes, and the lines are specified in this document for every point below. We hope we have met your requirements to improve this paper.
I am not sure: did you analyze only the main diagnosis in ICD or generally every number which is dedicated to the MI or stroke? Why do I ask- If you analyzed only the first given in-hospital number of ICD it could be confusing because if the patient with positive COVID suffered also from ischemia the main diagnosis ( so also code ICD) could be connected with COVID not: stroke or MI even if the patient had mild COVID but acute ischemic symptoms/sign. I am afraid that sometimes due to the payment (extra money was paid for COVID hospitalization but not for MI/stroke) the hospital authorities or just the practitioners could qualify the hospitalization as COVID, not MI/stroke. This could explain the lower rate (or pseudo-lower rate) of hospitalization in this "hot period" of lockdown. If, in your opinion, this is possible please state it as a limitation.
- We thank the reviewer for this comment. The ICD codes of myocardial infarction, stroke, and TIA were considered as a primary diagnosis but also as associated and secondary diagnoses to ensure that these diseases were identified even if another severe disease was the primary diagnosis. This precision has been added in the methods section line 103: “The codes were considered as a primary diagnosis but also as associated and secondary diagnoses to ensure that the four diseases were identified even if another severe disease was the primary diagnosis”.
The second possible limitation could be just the more frequent COVID-related death among the patients with the ischaemic disease ( in-home death) so they did not reach hospitalization.
- We agree with this comment. This point has been discussed line 337: “The more frequent COVID-related death among the patients with MI or stroke could also explain the decrease in hospital admissions for MI and stroke if the death occurred at home”.
In the literature, you can also find, that hospitalization depends on the age during COVID. Because the elderly were afraid about their health related to COVID and they "waited" with their symptoms/signs and do not visit the GP so, unfortunately, then they "visited" the emergency unit in the worst condition more frequently compared to the younger population (not because of the COVID but another disease)
Demczyszak I et al. The Use of Medical and Non-Medical Services by Older Inpatients from Emergency vs. Chronic Departments, during the SARS-CoV-2 Pandemic in Poland. Healthcare (Basel). 2021 Nov 12;9(11):1547. doi: 10.3390/healthcare9111547. PMID: 34828593; PMCID: PMC8624313.
- We agree with the reviewer and this point was discussed line 262: “Patients may have been afraid of exposure to COVID-19 in the hospital, especially the elderly, who may have stayed home with their symptoms without seeking medical attention, resulting in their admission to the emergency department in a more severe condition than the younger population”.

Reviewer 2 Report
.
Author Response
We thank the Reviewer for his positive comment.

Reviewer 3 Report
Thank you for giving me the opportunity to read and comment a report “Impact of the first COVID-19 wave on French hospitalizations for myocardial infarction and stroke: a retrospective cohort study.”, by Mariet A.S, et al.
In the reviewed manuscript, the volume of hospitalizations in France before and during the first lockdown in spring 2020 for MI, acute stroke and TIA has been investigated. Furthermore, the clinical 66 characteristics of patients with and without COVID-19 has been evaluated.
This paper is well written, correctly structured with a suitable research concept, the study limitations are addressed, and it is of relevance to readers of the journal. However, I include a few comments for your consideration.
· According to the journal's instructions, the abstract should be a total of about 200 words maximum. In this manuscript, the abstract is more than 300 words long. Please summarize it.
· The introduction section is very short, just a single paragraph, obviating the objectives. In the opinion of this reviewer, it would be necessary to describe in more depth the current state of the problem.
· Finally, it would be advisable to review the bibliography, since the references do not follow the format established by the journal.
Author Response
Comments and Suggestions for Authors
Thank you for giving me the opportunity to read and comment a report “Impact of the first COVID-19 wave on French hospitalizations for myocardial infarction and stroke: a retrospective cohort study.”, by Mariet A.S, et al.
In the reviewed manuscript, the volume of hospitalizations in France before and during the first lockdown in spring 2020 for MI, acute stroke and TIA has been investigated. Furthermore, the clinical 66 characteristics of patients with and without COVID-19 has been evaluated.
This paper is well written, correctly structured with a suitable research concept, the study limitations are addressed, and it is of relevance to readers of the journal. However, I include a few comments for your consideration.
- We thank the Reviewer for giving us an opportunity to substantially improve the content and the presentation of our manuscript. We have modified the article according to your requests. You will find the proposed modifications in the text using track changes, and the lines are specified in this document for every point below. We hope we have met your requirements to improve this paper.
- According to the journal's instructions, the abstract should be a total of about 200 words maximum. In this manuscript, the abstract is more than 300 words long. Please summarize it.
- We agree with the reviewer and the abstract has been reduced to 200 words.
- The introduction section is very short, just a single paragraph, obviating the objectives. In the opinion of this reviewer, it would be necessary to describe in more depth the current state of the problem.
- We agree with the reviewer and the introduction has been developed line 62 to 79.
- Finally, it would be advisable to review the bibliography, since the references do not follow the format established by the journal.
- The bibliography has been corrected according to the format established by the journal.

Round 2
Reviewer 1 Report
Ready for publication
Reviewer 3 Report
As far as this reviewer is concerned, the manuscript is suitable for publication.